# Implementation of Antimicrobial Stewardship Programs in Saudi Arabia: A Systematic Review

**DOI:** 10.3390/microorganisms13020440

**Published:** 2025-02-17

**Authors:** Abdullah A. Alshehri, Jehad A. Aldali, Maysoon A. Abdelhamid, Alaa A. Alanazi, Ratal B. Alhuraiz, Lamya Z. Alanazi, Meaad A. Alshmrani, Alhanouf M. Alqahtani, Maha I. Alrshoud, Reema F. Alharbi

**Affiliations:** 1Department of Clinical Pharmacy, College of Pharmacy, Taif University, Al Huwaya, Taif 21944, Saudi Arabia; a.aalshehri@tu.edu.sa; 2Department of Pathology, College of Medicine, Imam Mohammad Ibn Saud Islamic University (IMSIU), Riyadh 13317, Saudi Arabia; 3College of Medicine, Imam Mohammad Ibn Saud Islamic University (IMSIU), Riyadh 13317, Saudi Arabia; 443012915@sm.imamu.edu.sa (M.A.A.); 443012805@sm.imamu.edu.sa (A.A.A.); 443011511@sm.imamu.edu.sa (R.B.A.); 443013284@sm.imamu.edu.sa (L.Z.A.); 443013175@sm.imamu.edu.sa (M.A.A.); 443012622@sm.imamu.edu.sa (A.M.A.); 443021384@sm.imamu.edu.sa (M.I.A.); 443013084@sm.imamu.edu.sa (R.F.A.)

**Keywords:** antimicrobial stewardship programs, implementation, antimicrobial resistance, infection control, Saudi Arabia

## Abstract

Background: Antimicrobial resistance has highlighted the need for effective infectious disease strategies. Antimicrobial stewardship programs (ASPs) may reduce antibiotic resistance, adverse reactions, and treatment failures. This systematic review examines ASPs in Saudi Arabia, assessing their efficacy, challenges, and outcomes to improve antimicrobial use and patient care. Methods: Searches were carried out in the Ovid, MEDLINE, Embase, PsycInfo, Scopus, and Web of Science Core Collection databases for studies published from 2007 to July 2024, in Saudi Arabia, following the PRISMA guidelines. Studies that assessed ASPs’ implementation, effectiveness, and outcomes in hospital settings were included. Results: Out of the 6080 titles identified, 14 studies met the inclusion criteria, covering different regions of the country, including Riyadh, Jeddah, Dhahran, Makkah, Al-Kharj, and a multi-regional study in Qassim and Riyadh. Various interventions were implemented by the ASPs, such as educational programs, audit and feedback, switching from intravenous to oral administration, and enhanced policies. These interventions collectively led to a decrease in the overall antimicrobial consumption and cost, and a reduction in cases with multidrug-resistant bacteria. Conclusions: The findings of this review highlight the positive impact of ASPs in Saudi Arabia. However, addressing challenges such as data limitations and training gaps is essential to enhance their effectiveness. Expanding education and refining implementation strategies are crucial for ensuring their long-term success.

## 1. Introduction

A significant worldwide concern is antimicrobial resistance (AMR), which challenges our capacity to effectively stop and cure infectious diseases [1,2]. The various challenges of antimicrobial resistance, fueled by the various contributing factors, have given rise to a diverse array of complex problems affecting countries across the globe. Based on the available sources, these impacts can be categorized into three distinct domains: patient level, healthcare level, and economic level. Within the United States, antimicrobial resistance affects roughly two million individuals annually, leading to approximately 23,000 related fatalities. In 2019, the impact of AMR in Saudi Arabia was significant, with 2500 deaths directly attributed to AMR and an additional 9100 deaths associated with it [3,4]. Recent World Bank research indicated that the rise of antimicrobial resistance (AMR) will disproportionately affect low-income countries, potentially reducing the global GDP by about 1% annually, with developing nations facing a 5–7% loss by 2050. However, data on AMR’s mortality and economic impact in Saudi Arabia are lacking, underscoring the urgent need for national surveillance programs to address this issue. Patients with drug-resistant infections often require longer hospital stays and greater use of healthcare resources, such as ICUs and isolation beds, compared to those with non-resistant infections. This is due to the less effective combination treatments for resistant infections, which necessitate more intensive monitoring and containment measures [5].

Antimicrobial resistance arises from various factors, including healthcare practices, patient behaviors, and supply chain issues. The key contributors include inappropriate prescribing, insufficient patient education, limited diagnostic facilities, unauthorized sales, and weak regulatory mechanisms [6]. Also, the overuse of antimicrobials can contribute to the dissemination of drug-resistant microbes both locally and globally. A study in Makkah’s hospitals revealed that 61.9% of 710 hospitalized patients received one or more antimicrobial agents, indicating high antibiotic usage. This extensive consumption may increase the risk of developing and spreading resistant microbial strains [7].

A key strategy to combat antimicrobial resistance (AMR) is the implementation of antimicrobial stewardship programs (ASPs), which promote the responsible use of antimicrobials to reduce the development and spread of resistant microorganisms. Hospitals implement ASPs to mitigate antibiotic resistance, adverse reactions, treatment failures, and costs from prescriptions and extended stays. These programs focus on optimizing drug selection, dosage, and administration, monitoring usage and resistance patterns, and providing ongoing education and feedback to prescribers [2]. Recent advancements in artificial intelligence and machine learning, particularly self-supervised learning and multi-modality data integration, offer promising approaches for tracking and analyzing ASP outcomes [8,9,10]. Techniques such as adaptive deep learning frameworks, successfully applied in medical imaging and infectious disease detection, could enhance ASP monitoring by integrating diverse data sources, optimizing decision-making, and improving real-time responsiveness [11,12].

A survey of 247 MOH hospitals found a 54% response rate, with only 26% of responding hospitals implementing ASPs. Further evaluation is needed to assess the impact of ASP interventions on antimicrobial consumption and clinical outcomes in Saudi Arabia [13]. The barriers to and facilitators of adopting ASPs in Saudi Arabia are not well understood, with limited studies addressing the cultural, institutional, and resource-related challenges. Research is needed on the long-term effects of ASPs across various regions and their compliance with international and local standards. Addressing these gaps could help improve ASP adoption and sustainability in Saudi Arabia. To date, no systematic review has been conducted on this topic in the Saudi context. This study aims to systematically review and analyze the implementation of ASPs in Saudi Arabia, assessing their effectiveness, challenges, and outcomes to provide comprehensive insights and recommendations for optimizing antimicrobial use and improving patient care.

## 2. Methods

### 2.1. Protocol and Guidance

The study protocol was registered in PROSPERO (CRD42024576469) and followed the PRISMA, Cochrane Collaboration Handbook, and MOOSE (Meta-analysis of Observational Studies in Epidemiology) guidelines [14,15,16,17]. The PRISMA and MOOSE checklists are provided as in the Appendix A.

### 2.2. Search Strategies

A systematic search was conducted using Ovid, MEDLINE, Embase, PsycInfo, Scopus, and Web of Science Core Collection by AA, covering all the available literature until July 2024. The keywords included “Antimicrobial Stewardship Programs”, “Implementation”, “Antimicrobial Resistance”, “Antimicrobial Use”, “Infection Control”, and “Saudi Arabia”, combined with all the major regions in Saudi Arabia (Appendix A). Boolean operators (AND, OR) were used to refine the search results. The reference lists of included studies were manually screened for additional articles.

### 2.3. Inclusion Criteria

Eligible studies were conducted in Saudi Arabia and focused on implementing ASPs and assessing their effectiveness and outcomes in hospital settings for adults and children. The included designs were interventional, retrospective, prospective, quasi-experimental, cross-sectional, case reports, case series, and cohort studies in English.

### 2.4. Exclusion Criteria

The exclusions included review articles, opinion papers, studies on antimicrobial consumption alone, animal studies, and those with high risk of bias or poor data quality.

### 2.5. Study Selection and Data Extraction

The search results were exported to Rayyan QCRI for the systematic review screening. Duplicates were removed, and the titles and abstracts were screened by four groups of two reviewers (e.g., M.A. and R.A.). Conflicts were resolved by a third reviewer (A.A.). Data from the eligible studies were independently extracted into a standardized Excel sheet, summarizing study details such as the authors, publication year, design, location, setting, sample criteria, size, demographics, interventions, and outcomes.

### 2.6. Risk of Bias Assessment

The risk of bias was assessed using Joanna Briggs Institute (JBI) critical appraisal tools tailored to each study design [18]. Prospective and quasi-experimental studies were evaluated for the participant selection, intervention consistency, and follow-up completeness. Retrospective studies were assessed for the outcome measurement and confounding control, while qualitative studies were evaluated for the credibility and transferability. Two reviewers (e.g., M.A. and R.A.) independently applied the JBI checklists, with disagreements resolved by discussion or a third reviewer (A.A.).

### 2.7. Data Synthesis

A narrative synthesis was conducted, categorizing the studies by design, setting, and ASP interventions. The key themes included education, guideline implementation, audit and feedback, formulary restriction, IV to oral conversion, and antimicrobial consumption trends. The quantitative outcomes (e.g., antibiotic consumption, cost savings, resistance patterns) were summarized descriptively. Due to the study heterogeneity, meta-analysis was not possible.

## 3. Results

The initial database search retrieved 6080 titles and abstracts, and an additional paper was identified after reviewing the references and existing reviews. After further assessment, 5148 titles and abstracts were evaluated, and 35 studies were eligible for full-text review. A further 21 studies were excluded for the following reasons: different research aims (n = 6), full-text unavailable (n = 3), conducted outside Saudi Arabia (n = 4), review articles (n = 1), and focus on other antimicrobial stewardship aspects than hospital implementation (n = 7). One additional research paper was found through this manual search process and was included in the systematic review. Ultimately, 14 studies met the criteria for inclusion in the analysis. The study selection process is summarized in the flow diagram shown in Figure 1.

### 3.1. Study Characteristics

In Saudi Arabia, 14 studies investigated various AMS interventions across multiple cities and healthcare settings. Riyadh had the highest representation with four studies [19,20,21,22], followed by Jeddah with two studies [23,24] and Dhahran with two studies [25,26]. Makkah also had two studies [27,28], while Al-Kharj [29], Albaha [30], and South Saudi Arabia [31] each contributed one study. Additionally, one study covered both Riyadh and Qassim [32]. The predominant settings for these studies were hospitals, particularly tertiary care centers, intensive care units (ICUs) and military hospitals [19,20,21,22,23,24,25,26,27,28,29,30,31,32]. The characteristics of these studies are presented in Table 1.

A variety of study designs were employed, including six retrospective studies [19,20,21,22,24,26], three prospective interventional studies [23,25,29], two quasi-experimental studies [27,32], two cross-sectional studies [13,28], and one qualitative study [31].

The most common interventions reported across these studies included educational programs for healthcare professionals [19,22,23,24,25,27,28,32], the implementation of antimicrobial guidelines [22,23,27], and audit and feedback mechanisms to monitor and optimize antimicrobial use [19,22,24,25,27,30,32].

Several studies employed more specific interventions, such as intravenous (IV) to oral conversion protocols [19,25,27] and restrictions on broad-spectrum antibiotics, particularly carbapenems, to minimize resistance and improve targeted therapy [21,23,27,29]. Other strategies included hospital-wide antibiotic benchmarking [26] and perioperative prophylaxis adherence programs to reduce surgical site infections [29].

Additionally, some studies incorporated formulary restriction and preauthorization strategies to control high-risk antibiotic use, such as carbapenems [13,30], while others focused on antibiotic de-escalation and optimizing antibiotic dosing based on pharmacokinetic and pharmacodynamic (PK/PD) principles [25,27]. These interventions collectively aimed to enhance the rational use of antibiotics and reduce antimicrobial resistance in Saudi healthcare settings.

### 3.2. Reported Outcomes

To evaluate the interventions, the studies examined the consumption before and after antimicrobial stewardship program policy, cost savings, effects on multidrug resistance and susceptibility rates, and challenges faced. A summary of the included studies and their outcomes is provided in Table 2.

### 3.3. Studies Investigating ASP Interventions Used in Hospitals

Educational programs were used in most studies to address AMR. Alghamdi discussed an antibiotic use education program [15]. This annual infection control and antimicrobial stewardship symposium teaches clinicians how to use antibiotics properly. Another study described two important antimicrobial stewardship education programs: the first educates ICU specialists, residents, and pharmacists on infectious disease guidelines and stewardship tools; the second focuses on training hospital pharmacists [23]. Several studies involved educational sessions for healthcare providers [18,19,21,24,26,28]. A few articles targeted prescribers and pharmacists [21,28], while another reported common strategies like workshops, lectures, posters, and face-to-face interventions [24]. However, the lowest level of training through the ASP programs was offered to nursing staff, only 15.7%, compared to higher levels provided to physicians and pharmacists.

Several studies explored the impact of audit and feedback interventions on AMR. One study found that multidisciplinary ASP team feedback improved upper respiratory tract infection prescribing in pediatric outpatient clinics [21]. Similar findings were reported in studies where routine clinical rounds provided intensivists with ASP feedback [18,23]. Another study focused on prospective audit and feedback (PAAF) interventions, which included ASB cases under ASP oversight to enhance antimicrobial use [15].

A significant increase in IV to oral antibiotic switching, from 5.7% in 2016 to 31.3% in 2020, a 5.5-fold rise [19]. Another study found that 59% of antibiotics were given IV, 30.4% orally, and 10.6% were switched from IV to oral [15]. A significant decrease in IV antimicrobial use by 129 DDD per 100 bed-days (*p* = 0.038) but no significant change in oral antimicrobial use (*p* = 0.347) [23]. Examples of patients switched from IV to oral antibiotics were also documented [16]. In 2022, a study evaluated antibiotic prescriptions, focusing on the clinical suitability, dosage, duration, and guidelines [19]. Another report highlighted improvements in the antibiotic dosing, timing, route, and duration [25]. The development of evidence-based guidelines based on local resistance patterns [18]. Additionally, restrictions on broad-spectrum antibiotic prescribing were implemented [23].

### 3.4. Consumption, Cost Savings and Defined Daily Dose

Overall, the implementation of ASPs has led to a significant reduction in antibiotic consumption, as highlighted by several studies. One study noted that inappropriate antibiotic use decreased from 96.6% (53/55) to 94.4% (51/54) [15]. Another reported a 25% reduction in the average monthly antimicrobial utilization [23]. Additionally, a study observed a 67.2% decrease in dispensed restricted antibiotics, dropping from 17,763 units per month in 2012 to 5819 units in 2014, along with a nearly five-fold reduction in prescriptions, from 52,144 to 11,867 units per month during the same period [20].

Antimicrobial use, measured in DDDs per 1000 patient-days, increased before the launch of the AMS program but declined afterward [32]. In 2022, a study reported a decreasing trend in DDDs for most antibiotics from 2015 to 2019, except for meropenem, which rose from 26.1 to 79.9 between 2015 and 2016, and tigecycline, which peaked at 116.6 in 2017 before dropping to 3.21 in 2019 [19]. Another study observed a significant decline in carbapenem consumption, from 28.44 to 11.67 DDDs per 1000 patient-days (*p* = 0.012), while ciprofloxacin consumption increased, although not significantly (*p* = 0.190) [17]. Additionally, piperacillin/tazobactam consumption fell significantly from 16.46 to 5.89 DDDs per 1000 patient-days (*p* = 0.044). Another study reported total DDDs of 37,557 in 2013, 36,550 in 2014, and 38,738 in 2015, with the DDDs per 100 patient-days ranging from 90.7 to 94.5, and the days of therapy (DOT) rising from 35,218 in 2011 to 42,326 in 2015 [22]. Moreover, one study found that antibacterial use significantly decreased in the active ASP arm, with 376.2 DDDs per 1000 patient-days compared to 2403.64 DDDs per 1000 patient-days in the historical control arm, which had a baseline of 1177.8 DDDs [18].

Based on the information provided in the articles, the studies unanimously concluded that there was a notable decline in the overall cost of antibiotics over the examined period [16,18,19,20,25,28]. One cost analysis study and found a considerable decrease in hospital expenditure on restricted antibiotics from 2012 to 2014, with the average monthly savings being the highest in 2012 at SAR 1,222,613 (USD 326,020), followed by savings of SAR 488,834 (USD 130,350) in 2013 and SAR 391,594 (USD 104,420) in 2014 [20]. Similarly, another study reported a reduction in antibiotics utilization and direct costs, with the total cost of antibacterial agents during the study period decreasing from USD 760.37 in the historical control arm to USD 309.87 in the active antimicrobial stewardship program (ASP) arm [18].

### 3.5. Effect of Multidrug Resistance and Susceptibility Rate

The studies presented suggested that the implementation of various ASP interventions led to a decrease in multidrug resistance over time. One study observed an inverted U-shaped curve in terms of the multidrug-resistant healthcare-associated infections (MDR-HAIs) in intensive care units (ICUs) and long-term care (LTC) facilities, with an initial increase from 2016 to 2017, followed by a decrease to the lowest rates in 2020, highlighting reduced multidrug resistance after ASP implementation [19]. Similarly, another study reported a significant decrease in multidrug-resistant *Pseudomonas aeruginosa* isolates following carbapenem restriction measures [17]. Another study found that the proportion of multidrug-resistant isolates dropped from 44% before the intervention to 32.7% afterward, indicating a positive impact [15]. Additionally, an analysis of antimicrobial susceptibility patterns from 2017 to 2019 showed that the hospital maintained or slightly increased the microorganism susceptibility levels during this period [22]. One study also noted that after implementing a carbapenem restriction policy, the resistance of *Pseudomonas aeruginosa* to imipenem and meropenem significantly decreased, although the susceptibility to other antibacterial agents was not notably affected [26].

### 3.6. Challenges

Prior to implementing ASPs, one study reported that no data available for comparing the switch from intravenous (IV) to oral antibiotic administration, as this practice was not adopted earlier [19]. The initial phase of the ASP faced resistance from physicians, especially in the long-term care (LTC) unit, complicating adherence to the Electronic Antimicrobial Infection Use Guidelines (EAIUG). Another study noted that interventions were not implemented in a control group for ethical reasons, and some changes lacked statistical significance due to the limited number of gastrointestinal surgeries [25]. Issues with the electronic medical system’s compatibility with the ASP’s needs, requiring manual data generation, were also highlighted [26]. Regulatory restrictions hindered access to actual antibiotic purchase prices, and measuring antimicrobial consumption using the defined daily doses (DDDs) was problematic due to varying patient needs [23]. Additional studies identified significant challenges in managing multidrug-resistant (MDR) pathogens, including the complexity of carbapenem resistance and missing clinical outcome data [17,22]. Another study emphasized the need for more data on the reasons for de-escalation failure [16]. Other reported barriers included sociopolitical factors, organizational issues, healthcare professionals’ resistance, inadequate guideline enforcement, liability concerns, and insufficient resources, as described by [27]. Lastly, another study noted physicians’ resistance to new guidelines, suggesting the need for a complementary approach alongside restrictive methods [20].

### 3.7. Risk of Bias

The risk of bias across the studies in this systematic review, evaluated through the JBI criteria, is presented in Appendix A. The bias levels differed among the study types. Quasi-experimental studies showed robust compliance with the criteria for outcome reliability and statistical analysis. Cross-sectional studies adhered well to the criteria for inclusion clarity, exposure reliability, and statistical analysis, although some gaps were noted in addressing confounding factors. Qualitative studies exhibited mixed compliance, with challenges observed in accounting for the researcher influence and cultural context. Observational and retrospective cohort studies mostly met the key criteria but often lacked thorough strategies to manage confounding factors. Overall, the visual summary highlights the generally high compliance across the essential domains, with certain design aspects identified as potential areas for improvement.

## 4. Discussion

### 4.1. Summary of the Findings

This systematic review evaluates ASPs in Saudi Arabia, analyzing 14 studies that demonstrate improvements in antimicrobial use through educational programs, audit and feedback mechanisms, and IV to oral conversion protocols. While the initiatives primarily focused on physicians and pharmacists, training gaps for nursing staff were noted. Regular audits enhanced prescribing practices, especially in pediatric and ICU settings, leading to reduced antibiotic consumption, cost savings, and lower multidrug resistance. However, the barriers to broader ASP success included physician resistance, data limitations, and compatibility issues within healthcare systems.

### 4.2. Compared to the Literature Excellence

Optimal antimicrobial use is crucial for mitigating resistance and improving patient outcomes. The World Health Organization (WHO) advocated for global ASPs, highlighting their cost-effectiveness in reducing AMR and HAIs [33]. Studies showed that AMS interventions improve the quality and quantity of antimicrobial prescribing without compromising patient outcomes [34]. A Saudi survey found that only 26% of hospitals had ASPs, highlighting the need to evaluate their effects on antimicrobial use and clinical outcomes [13,28].

Numerous studies associate hospital ASP interventions with targeted educational strategies. Alghamdi highlighted an annual symposium on infection control and antimicrobial stewardship to improve clinicians’ understanding of appropriate antibiotic use. Similarly, Abdul Haseeb discussed two key educational programs: one for ICU specialists, residents, and pharmacists on infectious disease guidelines and ASP tools, and another for hospital pharmacists on the same topics [27]. Various researchers have conducted educational sessions and training for healthcare providers, focusing on prescribers and pharmacists [23,25,28,30]. Abdul Haseeb’s work suggested that workshops, lectures, posters, lunchtime talks, and face-to-face interventions are commonly employed strategies for educating doctors, pharmacists, and nurses, although nurses received significantly less ASP training compared to physicians and pharmacists. Additionally, several studies examined the audit and feedback effects on antimicrobial stewardship. Okeahialam found that audits and feedback improved antibiotic prescribing for pediatric respiratory infections. Similarly, Haseeb and Amer reported better antimicrobial use with feedback during clinical rounds. Alghamdi focused on prospective audit and feedback interventions to improve antimicrobial use across various conditions. Awad Al-Omari highlighted that both real-time and retrospective feedback are key components of ASPs [32].

The transition from intravenous (IV) to oral antibiotics has also seen considerable growth over time. Maha Alawi noted a marked rise in switching from IV to oral antibiotics during the study period. Alghamdi found that many antibiotics were initially given intravenously, with fewer converted to oral administration. Abdul Haseeb reported a significant reduction in the overall IV antimicrobial usage after the intervention, although the oral usage remained unchanged. Alshareef provided case studies demonstrating the growing adoption of this practice.

Studies have highlighted the growing adoption of IV to oral antibiotic conversion protocols. For instance, Alawi reported a 5.5-fold rise in IV to oral switches, Alghamdi noted 10.6% IV to oral conversions, and Haseeb observed reduced IV use with stable oral usage. Several studies reported efforts to optimize hospital antimicrobial policies. For instance, Maha Alawi reviewed antibiotic prescriptions for clinical appropriateness and guideline adherence. Nehad Ahmed highlighted improvements in dosage, timing, administration route, and prophylaxis duration. Marwa Amer noted evidence-based guidelines by an antimicrobial subcommittee, while Abdul Haseeb emphasized broad-spectrum antibiotic restrictions.

ASPs significantly reduced antibiotic consumption in many studies. Alghamdi noted a decline in inappropriate antibiotic use following the interventions, while Abdul Haseeb reported a significant reduction in the average monthly antimicrobial usage. Similarly, Maha M. Alawi observed a substantial decrease in the dispensation and prescription of restricted antibiotics. Awad Al-Omari found that antimicrobial use, measured in defined daily doses (DDDs) per patient-day, initially increased before the AMS program but declined after its implementation. Alawi’s research showed a decline in DDDs per patient-day for most antibiotics, with meropenem and tigecycline rising initially before decreasing. Abdallah reported reduced carbapenem use, with a non-significant rise in ciprofloxacin. Piperacillin/tazobactam usage significantly decreased. Hisham Momattin noted fluctuations in the total DDDs over the years, with the DDDs per patient-day remaining stable while the days of therapy increased. Marwa R. Amer’s research showed reduced antibacterial use in the ASP group versus a historical control.

Studies revealed that ASP interventions reduced multidrug resistance over time. For instance, Maha Mahmoud Alawi’s research noted a decline in MDR-HAIs in ICUs and long-term care facilities after ASP implementation, indicating a positive impact on MDR reduction. Mohammad Abdallah observed a significant decrease in multidrug-resistant *Pseudomonas aeruginosa* isolates following carbapenem restrictions. Similarly, Alghamdi reported a reduction in MDR isolates post-ASP intervention, reinforcing the positive effects of these programs on resistance patterns.

Hisham Momattin’s study showed stable or improved antimicrobial susceptibility over time. Additionally, Saleh Alghamdi’s research found carbapenem restrictions reduced *Pseudomonas aeruginosa* resistance, with other antibiotic susceptibilities remaining unchanged, while Saleh Alghamdi highlighted challenges like electronic system incompatibility with ASP needs, requiring manual data collection. Regulatory limitations hindered the accurate acquisition of antibiotic purchase prices, and using DDDs to measure antimicrobial consumption was challenging due to patient variability. Studies by Mohammad Abdallah and Hisham Momattin highlighted additional complexities, including carbapenem resistance, lack of clinical outcome data, incomplete susceptibility information, and potential confounding factors in managing multidrug-resistant pathogens.

Hanan Alshareef called for more data on de-escalation failures, while Saleh Alghamdi identified barriers to effective ASPs, such as sociopolitical challenges, resistance from healthcare professionals, poor guideline enforcement, liability concerns, and inadequate human and IT resources.

Studies have demonstrate varied ASP effects on antimicrobial use [18,28]. One study showed that ASPs improved the appropriateness of empirical antibiotic regimens in the ICU, although they did not address overall antibiotic volumes [18]. The other study found a significant and sustained reduction in antimicrobial costs post-ASP implementation [28]. Similarly, a study from Lebanon reported decreased broad-spectrum antibiotic use, lower prescribing rates, and improved infection outcomes [35]. These studies demonstrate the effectiveness of ASPs in improving antibiotic use, reducing costs, and lowering overall and specific antibiotic consumption.

The studies by Al-Tawfiq (2015) [36] and Sid Ahmed (2020) [9] demonstrated the impact of ASPs on susceptibility rates in different contexts. Al-Tawfiq reported mixed results, showing the increased susceptibility of *Enterobacter aerogenes* to amikacin and third-generation cephalosporins, while the susceptibility to piperacillin-tazobactam and ampicillin declined [36]. For *Pseudomonas aeruginosa*, the susceptibility to third-generation cephalosporins slightly improved, but the piperacillin-tazobactam susceptibility decreased. In contrast, Sid Ahmed observed a significant reduction in resistance among multidrug-resistant *Pseudomonas aeruginosa* to several antibiotics, indicating a more consistent decrease in resistance. Both studies highlighted the positive effects of ASPs, with Sid Ahmed’s findings showing broader reductions in antibiotic resistance.

Shaukat (2020) [37] and Khdour (2018) [38] demonstrated different approaches in their ASPs. Shaukat focused on reducing ceftriaxone use through intravenous to oral switches, clinician education, and substitution with penicillin, achieving decreased ceftriaxone consumption and shorter hospital stays. Khdour applied broader interventions like dosage optimization and de-escalation based on culture results, leading to significant reductions in the overall antimicrobial use and shorter therapy duration, with no notable changes in the 30-day mortality or readmission rates [38]. While Shaukat’s approach targeted ceftriaxone specifically, Khdour’s strategies offered general benefits across various antimicrobials and patient outcomes [37,38].

The findings of this systematic review provide valuable insights into the implementation and effectiveness of ASPs in Saudi Arabia. The results highlight key strategies such as educational interventions, formulary restrictions, IV to oral conversion protocols, and audit-feedback mechanisms, which have successfully reduced antibiotic consumption and improved adherence to guidelines. Given that antimicrobial resistance is a global concern, these findings may be applicable to other healthcare systems, particularly in countries with similar healthcare infrastructures, regulatory frameworks, and AMR challenges. Middle Eastern countries, as well as low- and middle-income countries facing high antibiotic overuse, could benefit from adopting similar ASP strategies to optimize antimicrobial prescribing.

This study has several limitations. First, limiting the review to English-language studies may have excluded relevant research published in other languages. This review excludes veterinary and agricultural ASPs. Given the One Health approach, future research should explore cross-sectoral AMR surveillance to strengthen stewardship efforts [39,40]. Methodological limitations introduced biases affecting reliability, as the cross-sectional studies lacked confounding adjustments, impacting the ASP effect estimates. The qualitative studies varied in rigor, affecting the consistency, while the retrospective and quasi-experimental studies had selection and measurement biases, limiting the causal inference. Additionally, publication bias may be present, as studies reporting negative or non-significant ASP outcomes could be under-represented. Furthermore, reporting bias could impact the findings, as some studies did not fully report key clinical outcomes such as patient safety and long-term resistance patterns. Finally, the variability in study designs, intervention types, and outcome measures introduced heterogeneity, restricting our ability to perform a meta-analysis and limiting the synthesis of the findings to a qualitative analysis.

Future research should prioritize longitudinal studies that evaluate ASP interventions over extended periods, providing more robust data on their impact on resistance patterns and patient outcomes. Additionally, exploring the role of digital health tools, such as electronic medical records and decision support systems, could enhance ASP implementation and monitoring. Adaptive deep learning and information fusion techniques could enhance ASPs by integrating prescribing trends, resistance patterns and surveillance data. This integration would enable real-time monitoring, outbreak prediction, and automated prescriber feedback to strengthen antimicrobial stewardship [41,42]. Comparative studies investigating ASPs across different healthcare settings, such as primary care versus tertiary care, would provide valuable insights into how ASPs can be tailored to various clinical environments. Finally, further research on the perspectives of healthcare professionals, especially nursing staff, would help address the educational gaps and optimize the effectiveness of ASP initiatives across interdisciplinary teams.

## 5. Conclusions

This systematic review highlights the positive outcomes of ASP implementation in Saudi Arabia, including reductions in antibiotic consumption, costs, and multidrug resistance. However, addressing challenges such as physician resistance, data limitations, and gaps in training is essential for further improving the effectiveness of ASPs. Ongoing efforts to expand education and refine implementation strategies are crucial to ensuring the continued success of antimicrobial stewardship programs.

## Figures and Tables

**Figure 1 microorganisms-13-00440-f001:**
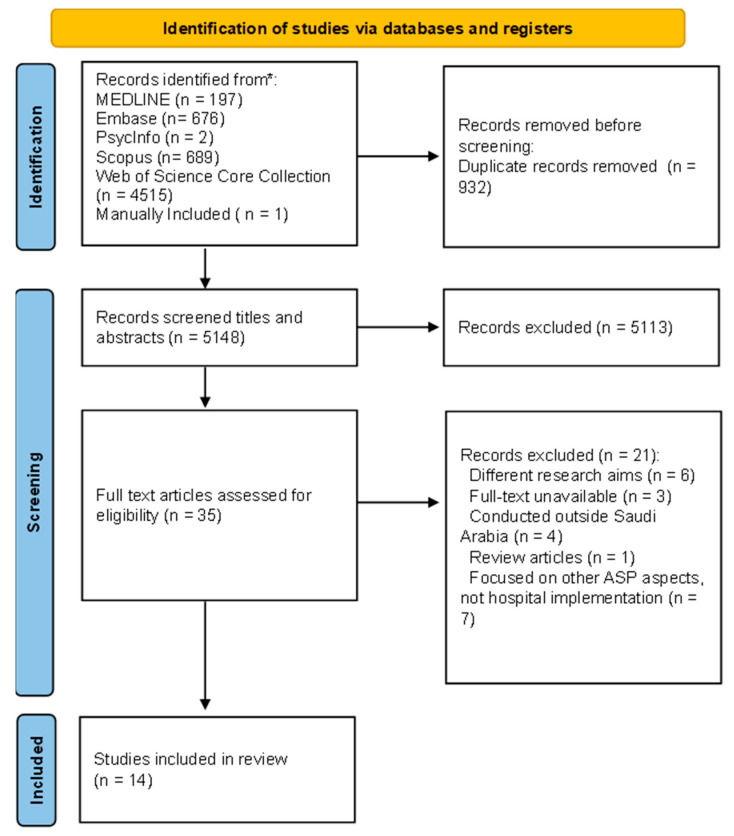
This diagram delineates the sequential procedures involved in the identification and selection of pertinent studies for a systematic review. Six electronic databases were searched: Ovid, MEDLINE, Embase, PsycInfo, Scopus, and Web of Science. Fourteen studies fulfilled the inclusion criteria. * Reporting the number of records identified from each database or register searched (rather than the total number across all databases/registers).

**Table 1 microorganisms-13-00440-t001:** Characteristics of the included studies. N/A: Not Applicable.

Authors and Year	City of Study	Setting	Study Design	Interventions Details	Reported Outcomes
Ahlam Alghamdi (2021) [15]	Riyadh	Hospital	Retrospective study	Education programs, antimicrobial guidelines, carbapenem restriction and prior authorization, IV to oral conversion, prospective audit and feedback	Enhanced adherence to bacteriuria management guidelines and reduced inappropriate antibiotic use
Hanan Alshareef (2020) [16]	Riyadh	Hospital	Retrospective cohort study	Implementation of antibiotic de-escalation strategies in UTI patients, clinical outcome monitoring	Better patient outcomes, decreased resistance rates, and shorter antibiotic therapy duration
Mohammed Abdallah (2017) [17]	Riyadh	ICU	Retrospective study	Restriction of broad-spectrum antibiotics in ICU settings	Enhanced *Pseudomonas aeruginosa* susceptibility following carbapenem restriction
Marwa R. Amer (2013) [18]	Riyadh	ICU	Comparative controlled study	Comprehensive ASP implementation, structured educational programs, prescriber training, antibiotic use audits and restriction policies	Lower antibiotic consumption and reduced antimicrobial resistance in ICU patients
Maha M. Alawi (2022) [19]	Jeddah	Long-term care facility	Prospective interventional study	Educational/advisory role, Anti-infective use guidelines	Decreased infection rates, reduced antimicrobial use, and improved stewardship adherence
Maha M. Alawi (2016) [20]	Jeddah	Tertiary hospital	Retrospective study	Stepwise ASP introduction, education programs, antimicrobial prescribing guidelines, audit–feedback mechanisms	Increased compliance with ASP guidelines and reduced inappropriate antibiotic prescribing
Christopher A. Okeahialam (2021) [21]	Dhahran	Hospital	Retrospective study	Educational sessions, prospective audits and feedback, pre-operative antibiotic protocols, automatic renal dosing, IV to oral conversion	Decreased *C. difficile* infection rates, supporting its role in ASP evaluation
Hisham Momattin (2018) [22]	Dhahran	Hospital	Retrospective study	Hospital-wide antibiotic benchmarking	More effective ASP monitoring and improved prescribing practices
Abdul Haseeb (2021) [23]	Makkah	Critical care unit	Quasi-experimental study	Education for healthcare teams, broad-spectrum antibiotic restriction, optimized antibiotic dosing (PK/PD analysis), regular audits and clinician feedback, chart stickers for IV to oral conversion	Optimized antibiotic therapy, lower resistance rates, and improved patient outcomes
Abdul Haseeb (2020) [24]	Makkah	Multiple hospitals	Cross-sectional study	Antimicrobial stewardship education via workshops, lectures, posters, and face-to-face interventions	Identified ASP success levels and barriers to implementation across hospitals
Nehad J. Ahmed (2022) [25]	Al-Kharj	Perioperative care	Interventional study	ASP for perioperative prophylaxis, surgical team workshops, treatment protocol dissemination, compliance monitoring	Greater adherence to prophylaxis guidelines and reduced surgical site infections
Saleh Alghamdi (2021) [26]	Albaha	Medical city	Case study	Restriction and preauthorization, prospective audit and intervention feedback	Significant reduction in inappropriate prescriptions and improved antibiotic regulation
Saleh Alghamdi (2019) [27]	South Saudi Arabia	Three hospitals	Qualitative study	N/A	Major ASP barriers identified, including lack of engagement from medical staff and insufficient resources
Awad Al-Omari (2020) [28]	Qassim and Riyadh	Four tertiary hospitals	Pre-post quasi-experimental study	Education for prescribers and healthcare workers, audit and feedback (real time/retrospective)	Lower antibiotic consumption, significant resistance decline, and improved patient safety

**Table 2 microorganisms-13-00440-t002:** A summary of the included studies and their outcomes for ASPs in Saudi Arabia.

Authors and Year	Outcome(s) Assessed	Key Results
Ahlam Alghamdi (2021) [15]	Inappropriate antibiotic use in asymptomatic bacteriuria (ASB) before and after ASP implementation	-Inappropriate antibiotic use in ASB before ASP: 95%. -Reduction in carbapenem use: (*p* = 0.04). -Increase in cephalosporin use: (*p* = 0.01). -MDR bacteria prevalence: 38.7%.
Hanan Alshareef (2020) [16]	Antibiotic de-escalation impact on UTI patients	-De-escalation rate: 29.7% (27/91). -Median hospital LOS: 3 days (IQR: 2–6) in de-escalated vs. 10 days (IQR: 6–21) in failed de-escalation (*p* < 0.001). -MDR pathogens in failed group: 59.4% vs. 22.2% in de-escalated group (*p* < 0.001).
Mohammed Abdallah (2017) [17]	Impact of carbapenem restriction on *Pseudomonas aeruginosa* susceptibility	-Carbapenem consumption reduced: 28.44 → 11.67 DDDs/1000 patient-days (*p* = 0.012). -Resistance to Imipenem reduced: 76.0% → 38.5% (*p* = 0.019).
Marwa R. Amer (2013) [18]	ASP impact on antibiotic consumption, prescribing appropriateness, and ICU outcomes	-Appropriateness of empirical antibiotic therapy improved: 30.6% → 100% (*p* < 0.05). -Reduction in antibiotic utilization: 2403.64 → 376.2 DDDs/1000 patient-days. -ICU mortality rate decreased: 32.65% → 16.7%.
Maha M. Alawi (2022) [19]	Long-term ASP impact on infection control and antimicrobial resistance	-IV to oral switch rate increased: 5.7% → 31.3%. -Restricted IV antimicrobial consumption decreased by 40%. -MDR-HAI rate in ICU decreased: 3.22 → 1.14 per 1000 patient-days.
Maha M. Alawi (2016) [20]	Stepwise implementation of ASP in a tertiary care hospital	-Restricted antibiotic prescriptions decreased by 75%. -Significant reduction in *Acinetobacter baumannii* infections (*p* = 0.007).
Christopher A. Okeahialam (2021) [21]	*C. difficile* infections as an ASP performance indicator	-Toxigenic *C. difficile* positivity rate decreased: 9.8% → 7.4% (*p* = 0.022). -Significant decline in pediatric cases: 10.8% → 2.3% (*p* = 0.002).
Hisham Momattin (2018) [22]	Benchmarking antibiotic usage to reflect ASP outcomes	-Total DDD remained stable: 37,557 (2013) → 38,738 (2015). -Reduction in antibiotic usage (after adjustment for 100 bed-days and CMI): 5.13% decrease.
Abdul Haseeb (2021) [23]	Evaluation of a multidisciplinary ASP in a Saudi critical care unit	-Total antimicrobial consumption decreased: 742.86 → 555.33 DDD/100 bed-days (*p* = 0.110). -Significant reduction in IV antimicrobial usage: 129 DDD/100 bed-days (*p* = 0.038).
Abdul Haseeb (2020) [24]	Perceived success of ASPs in Makkah-region hospitals	-Hospitals surveyed 25; response rate: 76% (19 hospitals). -Most common ASP strategies: Formulary restrictions (89%), drug and therapeutics committee involvement (79%).
Nehad J. Ahmed (2022) [25]	Implementation of an ASP to improve adherence to perioperative prophylaxis guidelines	-Guideline adherence improved: 60% → 92%. -Surgical site infection (SSI) rate decreased: 0.41% → 0.04%.
Saleh Alghamdi (2021) [26]	Implementation of ASP in a Saudi medical city: Barriers and outcomes	-Implementation barriers identified: Lack of expertise, IT system incompatibility, resistance from physicians.
Saleh Alghamdi (2019) [27]	Barriers to implementing ASPs in three Saudi hospitals	-Identified key barriers to ASP adoption and implementation: -Lack of adherence to guidelines and enforcement. -Poor communication and team disintegration. -Shortage of ASP team members.
Awad Al-Omari (2020) [28]	Impact of AMS on antibiotic use, cost, and HAIs in four tertiary hospitals	-Antibiotic consumption reduced: 320 → 233 DDDs/1000 patient-days (*p* = 0.689). -Cumulative cost savings: SAR 6,286,929.

Abbreviations: AMS—antimicrobial stewardship, ASP—antimicrobial stewardship program, ASB—asymptomatic bacteriuria, CMI—case mix index, DDD—defined daily dose, HAI—healthcare-associated infection, ICU—intensive care unit, IQR—interquartile range, IV—intravenous, LOS—length of stay, MDR—multidrug-resistant, MDR-HAI—multidrug-resistant healthcare-associated infection, SAR—Saudi Riyal, SSI—surgical site infection, UTI—urinary tract infection.

## Data Availability

All data analyzed in this study were obtained from previously published sources, which are ap-propriately cited in the manuscript.

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
