# Peer review of "Implementation of Antimicrobial Stewardship Programs in Saudi Arabia: A Systematic Review"

_microorganisms, 2025, doi:10.3390/microorganisms13020440_

Round 1
Reviewer 1 Report
Comments and Suggestions for Authors
Dear authors,
I have now completed the review of the manuscript titled "Implementation of Antimicrobial Stewardship Programs in Saudi Arabia: A Systematic Review."
The overall study makes a valuable contribution to the science. Also, the manuscript is interesting and, in general, fairly well-written.
However, I still have some suggestions to further improve the quality of the manuscript.
1. The risk of bias assessment using JBI tools reveals concerning methodological weaknesses across the included studies. The authors acknowledge that cross-sectional studies had gaps in addressing confounding factors, and qualitative studies showed mixed compliance with quality criteria. However, they do not sufficiently discuss how these quality limitations might impact the reliability of their findings.
2. Introduction does not introduce articles which could be adapted for tracking and analyzing ASP outcomes. For example, Recent Deep Learning-Based Brain Tumor Segmentation Models Using Multi-Modality Magnetic Resonance Imaging: A Prospective Survey would be valuable because it demonstrates advanced methodologies for analyzing medical data. The multi-modality approach could inspire better ways to integrate different data sources in ASP monitoring. Also, Consequential Advancements of Self-Supervised Learning (SSL) in Deep Learning Contexts would be helpful for understanding modern data analysis approaches that could improve ASP implementation and monitoring. Self-supervised learning techniques could be particularly useful for analyzing patterns in antimicrobial usage and resistance.
3. The review's scope could be broadened. The exclusion of veterinary and agricultural antimicrobial stewardship programs limits the comprehensive understanding of AMR control in Saudi Arabia, given the One Health approach recommended for addressing antimicrobial resistance. In Deep Network Selection and Optimized Information Fusion for Enhanced COVID-19 Detection, the article would provide insights into how to better integrate multiple data sources and optimize decision-making processes in ASP implementation. The information fusion techniques could be particularly relevant for combining different aspects of antimicrobial stewardship data.
4. While the authors provide suggestions for future research, their recommendations could be more specific. For example, discussing ones like An Adaptive Ensemble Deep Learning Framework for Reliable Detection of Pandemic Patients would be valuable because it presents frameworks for detecting and managing infectious diseases, which directly relates to ASP implementation. The adaptive framework approach could be applied to improving ASP responsiveness.
Thank you for your valuable contributions to our field of research. I look forward to receiving the revised manuscript.
Reviewer 2 Report
Comments and Suggestions for Authors
Comments to authors
The authors conducted a systematic review of antimicrobial stewardship programmes in Saudi Arabia and their impact on several outcomes of interest. Given the current problem of multidrug resistance, the study is of interest. However, it needs some methodological improvements before its hypothetical publication. Furthermore, and more importantly, it should be clarified whether the findings of this review are applicable to other populations.
Abstract
· In the methods section, it is still necessary to define which studies were included (i.e. the inclusion criteria, "studies that ..... were included").
· The conclusion could be summarised in half.
Introduction
The introduction is coherent and complete.
Methods
· 2.1. Protocol and guidance”: In addition to PRISMA, authors should follow the Cochrane Collaboration Handbook and MOOSE guidelines.
· In addition, authors should add the PRISMA and MOOSE Checklists in supplementary material.
· “2.2. Search strategies”: Authors should describe the search strategy used. However, the authors may have included the search strategy in supplementary material, but I cannot access it.
· Authors should not specify a lower date range for the search. This means that the search should start from the beginning.
· The authors provide a number of appropriate inclusion criteria. However, the proposed exclusion criteria are basically the opposite of the inclusion criteria. In reality, exclusion criteria should be those that exclude studies (or participants) even if they meet the inclusion criteria.
· Authors should provide a citation for the JBI tool.
· There is a missing section before the results, entitled "Data synthesis", where the details of how the extracted data were processed are described (if there was a meta-analysis, it would also describe how the meta-analysis was performed, but this is not the case).
Results
· Figure 1: Records excluded (n=21) and Full-text articles excluded with reasons. Please, this should be in one box.
· If Medline gives only 197 results and WoS gives 4515, check that the search has been carried out correctly. In WoS, you should only search in the specific WoS database (WoS gives access to searches in other databases and an error could result in all databases being searched). Also, the search in WoS should be similar to the search in PubMed. Finally, I suggest not to use EMBASE or Scopus. In fact, these two databases are closely related.
· "3.1. Study characteristics": the authors give the name of the author, the year of publication and the citation. To reduce space and make reading easier, only the citation should be included.
· "All results are summarized in Table 1": However, Table 1 does not report the results. I think it would be appropriate to create a Table 2 that summarises the results of each study for each outcome. The results section is indeed of great interest, but Table 2 of results is missing.
· Figure 2: better in the supplementary material.
Discussion
· Before the limitations section, a paragraph describing the clinical implications of your article should be included. It should also discuss whether your results are extensible to other countries (I assume they are, but since it is a systematic review from Saudi Arabia only, a reader from another country will be interested in knowing whether your results are extensible).
· Finally, the limitations paragraph should be expanded a bit. In particular, I think that some biases, which may well exist, should be acknowledged.
References
Ok
Round 2
Reviewer 1 Report
Comments and Suggestions for Authors
All comments have been thoroughly addressed. I extend my gratitude to both the authors and editors for taking my opinions into consideration during the review of this manuscript.
Reviewer 2 Report
Comments and Suggestions for Authors
Comments to authors
The authors have done a good job of addressing the comments. However, I suggest that they address two comments that I think could be improved.
- Authors should provide one citation for PRISMA, one for the Cochrane Collaboration Handbook and one for MOOSE.
- Authors should provide the main reason for exclusion for each of the 21 excluded trials (AMSTAR-2 requires this).
